# Multi-view Matrix Factorization for Linear Dynamical System Estimation

**Mahdi Karami, Martha White, Dale Schuurmans, Csaba Szepesvári**
Department of Computer Science
University of Alberta
Edmonton, AB, Canada
`{karami1, whitem, daes, szepesva}@ualberta.ca`

## Abstract

We consider maximum likelihood estimation of linear dynamical systems with generalized-linear observation models. Maximum likelihood is typically considered to be hard in this setting since latent states and transition parameters must be inferred jointly. Given that expectation-maximization does not scale and is prone to local minima, moment-matching approaches from the subspace identification literature have become standard, despite known statistical efficiency issues. In this paper, we instead reconsider likelihood maximization and develop an optimization based strategy for recovering the latent states and transition parameters. Key to the approach is a two-view reformulation of maximum likelihood estimation for linear dynamical systems that enables the use of global optimization algorithms for matrix factorization. We show that the proposed estimation strategy outperforms widely-used identification algorithms such as subspace identification methods, both in terms of accuracy and runtime.

## 1 Introduction

Linear dynamical systems (LDS) provide a fundamental model for estimation and forecasting in discrete-time multi-variate time series. In an LDS, each observation is associated with a latent state; these unobserved states evolve as a Gauss-Markov process where each state is a linear function of the previous state plus noise. Such a model of a partially observed dynamical system has been widely adopted, particularly due to its efficiency for prediction of future observations using Kalman filtering.

Estimating the parameters of an LDS—sometimes referred to as system identification—is a difficult problem, particularly if the goal is to obtain the maximum likelihood estimate of parameters. Consequently, spectral methods from the subspace identification literature, based on moment-matching rather than maximum likelihood, have become popular. These methods provide closed form solutions, often involving a singular value decomposition of a matrix constructed from the empirical moments of observations (Moonen and Ramos, 1993; Van Overschee and De Moor, 1994; Viberg, 1995; Katayama, 2006; Song et al., 2010; Boots and Gordon, 2012). The most widely used such algorithms for parameter estimation in LDSs are the family of N4SID algorithms (Van Overschee and De Moor, 1994), which are computationally efficient and asymptotically consistent (Andersson, 2009; Hsu et al., 2012). Recent evidence, however, suggests that these moment-matching approaches may suffer from weak statistical efficiency, performing particularly poorly with small sample sizes (Foster et al., 2012; Zhao and Poupart, 2014).

Maximum likelihood for LDS estimation, on the other hand, has several advantages. For example, it is asymptotically efficient under general conditions (Cramér, 1946, Ch.33), and this property often translates to near-minimax finite-sample performance. Further, maximum likelihood is amenable to coping with missing data. Another benefit is that, since the likelihood for exponential families

and corresponding convex losses (Bregman divergences) are well understood (Banerjee et al., 2005), maximum likelihood approaches can generalize to a broad range of distributions over the observations. Similarly, other common machine learning techniques, such as regularization, can be naturally incorporated in a maximum likelihood framework, interpretable as maximum a posteriori estimation.

Unfortunately, unlike spectral methods, there is no known efficient algorithm for recovering parameters that maximize the marginal likelihood of observed data in an LDS. Standard iterative approaches are based on EM (Ghahramani and Hinton, 1996; Roweis and Ghahramani, 1999), which are computationally expensive and have been observed to produce locally optimal solutions that yield poor results (Katayama, 2006). A classical system identification method, called the prediction error method (PEM), is based on minimization of prediction error and can be interpreted as maximum likelihood estimation under certain distributional assumptions (e.g., Ch. 7.4 of Ljung 1999, Åström 1980). PEM, however, is prone to local minima and requires selection of a canonical parameterization, which can be difficult in practice and can result in ill-conditioned problems (Katayama, 2006).

In this paper, we propose an alternative approach to LDS parameter estimation under exponential family observation noise. In particular, we reformulate the LDS as a two-view generative model, which allows us to approximate the estimation task as a form of matrix factorization, and apply recent global optimization techniques for such models (Zhang et al., 2012; Yu et al., 2014). To extend these previous algorithms to this setting, we provide a novel proximal update for the two-view approach that significantly simplifies the algorithm. Finally, for forecasting on synthetic and real data, we demonstrate that the proposed algorithm matches or outperforms N4SID, while scaling better with increasing sample size and data dimension.

## 2  Linear dynamical systems

We address discrete-time, time-invariant linear dynamical systems, specified as

$$\begin{aligned} \phi_{t+1} &= \mathbf{A}\phi_t + \eta_t \\ \mathbf{x}_t &= \mathbf{C}\phi_t + \epsilon_t \end{aligned} \tag{1}$$

where $\phi_t \in \mathbb{R}^k$ is the hidden state at time $t$; $\mathbf{x}_t \in \mathbb{R}^d$ is the observation vector at time $t$; $\mathbf{A} \in \mathbb{R}^{k \times k}$ is the dynamics matrix; $\mathbf{C} \in \mathbb{R}^{d \times k}$ is the observation matrix; $\eta$ is the state evolution noise; and $\epsilon$ is the observation noise. The noise terms are assumed to be independent. As is common, we assume that the state evolution noise is Gaussian: $\eta \sim \mathcal{N}(\mathbf{0}, \mathbf{\Sigma}_\eta)$. We additionally allow for general observation noise to be generated from an exponential family distribution (e.g., Poisson). The graphical representation for this LDS is shown in Figure 1.

An LDS encodes the intuition that a latent state is driving the dynamics, which can significantly simplify estimation and forecasting. The observations typically contain only partial information about the environment (such as in the form of limited sensors), and further may contain noisy or even irrelevant observations. Learning transition models for such observations can be complex, particularly if the observations are high-dimensional. For example, in spatiotemporal processes, the data is typically extremely high-dimensional, composed of structured grid data; however, it is possible to extract a low-rank state-space that significantly simplifies analysis (Gelfand et al., 2010, Chapter 8). Further, for forecasting, iterating transitions for such a low-rank state-space can provide longer range predictions with less error accumulation than iterating with the observations themselves.

The estimation problem for an LDS involves extracting the unknown parameters, given a time series of observations $\mathbf{x}_1, \ldots, \mathbf{x}_T$. Unfortunately, jointly estimating the parameters $\mathbf{A}, \mathbf{C}$ and $\phi_t$ is difficult because the multiplication of these variables typically results in a nonconvex optimization. Given the latent states $\phi_t$, estimation of $\mathbf{A}$ and $\mathbf{C}$ is more straightforward, though there are still some issues with maintaining stability (Siddiqi et al., 2007). There are some recent advances improving estimation in time series models using matrix factorization. White et al. (2015) provide a convex formulation for auto-regressive moving average models—although related to state-space models, these do not permit a straightforward conversion between the parameters of one to the other. Yu et al. (2015) factorize the observation into a hidden state and dictionary, using a temporal regularizer on the extracted hidden state—the resulting algorithm, however, is not guaranteed to provide an optimal solution.

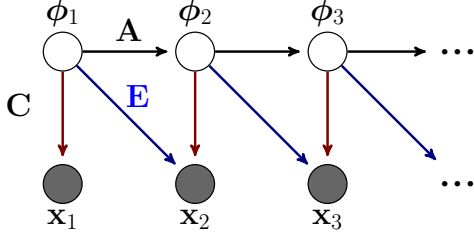

Figure 1: Graphical representation for the standard LDS formulation and the corresponding two-view model. The two-view formulation is obtained by a linear transformation of the LDS model. The LDS model includes only parameters $\mathbf{C}$ and $\mathbf{A}$ and the two-view model includes parameters $\mathbf{C}$ and $\mathbf{E} = \mathbf{CA}$, where $\mathbf{A}$ can be extracted from $\mathbf{E}$ after $\mathbf{C}$ and $\mathbf{E}$ are estimated.

## 3 Two-view Formulation of LDS

In this section, we reformulate the LDS as a generative two-view model with a shared latent factor. In the following section, we demonstrate how to estimate the parameters of this reformulation optimally, from which parameter estimates of the original LDS can be recovered.

To obtain a two-view formulation, we re-express the two equations for the LDS as two equations for pairs of sequential observations. To do so, we multiply the state evolution equation in (1) by $\mathbf{C}$ and add $\epsilon_{t+1}$ to obtain $\mathbf{C}\phi_{t+1} + \epsilon_{t+1} = \mathbf{CA}\phi_t + \mathbf{C}\eta_t + \epsilon_{t+1}$; representing the LDS model as

$$
\begin{aligned}
\mathbf{x}_{t+1} &= \mathbf{E}\phi_t + \epsilon'_{t+1} \\
\mathbf{x}_t &= \mathbf{C}\phi_t + \epsilon_t
\end{aligned}
\tag{2}
$$

where we refer to $\mathbf{E} := \mathbf{CA}$ as the factor loading matrix and $\epsilon'_{t+1} := \mathbf{C}\eta_t + \epsilon_{t+1}$ as the noise of the second view. We then have a two-view problem where we need to estimate parameters $\mathbf{E}$ and $\mathbf{C}$. Since the noise components $\epsilon_t$ and $\epsilon'_t$ are independent, the two views $\mathbf{x}_t$ and $\mathbf{x}_{t+1}$ are conditionally independent given the shared latent state $\phi_t$. The maximum log likelihood problem for the two-view formulation then becomes

$$
\max_{\mathbf{C},\mathbf{E},\Phi} \log p(\mathbf{x}_1,\ldots,\mathbf{x}_T|\phi_0,\phi_1,\ldots,\phi_T,\mathbf{C},\mathbf{E}) = \max_{\mathbf{C},\mathbf{E},\Phi} \sum_{t=1}^{T} \log p(\mathbf{x}_t|\phi_{t-1},\phi_t,\mathbf{C},\mathbf{E})
\tag{3}
$$

where, given the hidden states, the observations are conditionally independent. The log-likelihood (3) is equivalent to the original LDS, but is expressed in terms of the distribution $p(\mathbf{x}_t|\phi_{t-1},\phi_t,\mathbf{C},\mathbf{E})$, where the probability of an observation increases if it has high probability under both $\phi_{t-1}$ and $\phi_t$. The graphical depiction of the LDS and its implied two-view model is illustrated in Figure 1.

### 3.1 Relaxation

To tackle the estimation problem, we reformulate the estimation problem for this equivalent two-view model of the LDS. Note that according to the two-view model (2), the conditional distribution (3) can be expressed as $p(\mathbf{x}_t|\phi_{t-1},\phi_t,\mathbf{C},\mathbf{E}) = p(\mathbf{x}_t|\mathbf{E}\phi_{t-1}) = p(\mathbf{x}_t|\mathbf{C}\phi_t)$. Substituting each of these in the summation (3) would result in a factor loading model that ignores the temporal correlation among data; therefore, to take the system dynamics into account we choose a balanced averaging of both as $\log p(\mathbf{x}_t|\phi_{t-1},\phi_t,\mathbf{C},\mathbf{E}) = \frac{1}{2}\log p(\mathbf{x}_t|\mathbf{E}\phi_{t-1}) + \frac{1}{2}\log p(\mathbf{x}_t|\mathbf{C}\phi_t)$, where the likelihood of an observation increases if it has high conditional likelihood given both $\phi_{t-1}$ and $\phi_t$.[1] With this choice and the exponential family specified by the log-normalizer (also called potential function) $F : \mathbb{R}^d \to \mathbb{R}$, with the corresponding Bregman divergence defined as $D_F(\hat{\mathbf{z}}\|\mathbf{z}) := F(\hat{\mathbf{z}}) - F(\mathbf{z}) - \mathbf{f}(\mathbf{z})^\top(\hat{\mathbf{z}} - \mathbf{z})$ using transfer function $\mathbf{f} = \nabla F$,[2] the log-likelihood separates into the two components

$$
\underset{\mathbf{C},\mathbf{E},\Phi}{\operatorname{argmax}} \sum_{t=1}^{T} \log p(\mathbf{x}_t|\phi_{t-1},\phi_t,\mathbf{C},\mathbf{E}) = \underset{\mathbf{C},\mathbf{E},\Phi}{\operatorname{argmax}} \frac{1}{2} \sum_{t=1}^{T} \log p(\mathbf{x}_t|\mathbf{E}\phi_{t-1}) + \log p(\mathbf{x}_t|\mathbf{C}\phi_t)
$$

$$
= \underset{\mathbf{C},\mathbf{E},\Phi}{\operatorname{argmin}} \sum_{t=1}^{T} D_F(\mathbf{E}\phi_{t-1}\|f^{-1}(\mathbf{x}_t)) + D_F(\mathbf{C}\phi_t\|f^{-1}(\mathbf{x}_t))
$$

Each Bregman divergence term can be interpreted as the fitness measure for each view. For example, a Gaussian distribution can be expressed by an exponential family defined by $F(\mathbf{z}) = \frac{1}{2}\|\mathbf{z}\|_2^2$. The above derivation could be extended to different variance terms for $\epsilon$ and $\epsilon'$, which would result in different weights on the two Bregman divergences above. Further, we could also allow different exponential families (hence different Bregman divergences) for the two distributions; however, there is no clear reason why this would be beneficial over simply selecting the same exponential family, since both describe $\mathbf{x}_t$. In this work, therefore, we will explore a balanced loss, with the same exponential family for each view.

In order to obtain a low rank solution, one can relax the hard rank constraint and employ the block norm $\|\mathbf{\Phi}\|_{2,1} = \sum_{j=1}^k \|\mathbf{\Phi}_{j:}\|_2$ as the rank-reducing regularizer on the latent state.[3] This regularizer offers an adaptive rank reducing scheme that zeros out many of the rows of the latent states and hence results a low rank solution without knowing the rank *a priori*. For the reconstruction models $\mathbf{C}$ and $\mathbf{E}$, we need to specify a prior that respects the conditional independence of the views $\mathbf{x}_t$ and $\mathbf{x}_{t+1}$ given $\phi_t$. This goal can be achieved if $\mathbf{C}$ and $\mathbf{E}$ are constrained individually so that they do not compete against each other to reconstruct their respective views (White et al., 2012). Incorporating the regularizer and constraints, the resulting optimization problem has the form

$$\underset{\mathbf{C},\mathbf{E},\mathbf{\Phi}}{\operatorname{argmin}} \sum_{t=1}^T \mathcal{L}_1(\mathbf{E}\phi_{t-1}; \mathbf{x}_t) + \mathcal{L}_2(\mathbf{C}\phi_t; \mathbf{x}_t) + \lambda \sum_{j=1}^k \|\mathbf{\Phi}_{j:}\|_2 \qquad (4)$$

$$\text{s.t.} \|\mathbf{C}_{:j}\|_2 \leq \gamma_1, \|\mathbf{E}_{:j}\|_2 \leq \gamma_2 \ \forall j \in (1, k).$$

The above constrained optimization problem is convex in each of the factor loading matrices $\{\mathbf{C}, \mathbf{E}\}$ and the state matrix $\mathbf{\Phi}$, but not jointly convex in terms of all these variables. Nevertheless, the following lemma show that (4) admits a convex reformulation by change of variable.

**Lemma 1** *Let* $\hat{\mathbf{Z}}^{(1)} := \mathbf{C}\mathbf{\Phi}$ *and* $\hat{\mathbf{Z}}^{(2)} := \mathbf{E}\mathbf{\Phi}$ *with their concatenated matrix* $\hat{\mathbf{Z}} := \begin{bmatrix} \hat{\mathbf{Z}}^{(1)} \\ \hat{\mathbf{Z}}^{(2)} \end{bmatrix}$ *and*

$\mathbf{Z}^{(1)} := [\mathbf{x}_{1:T-1}]$, $\mathbf{Z}^{(2)} := [\mathbf{x}_{2:T}]$. *In addition, let's define* $\mathbf{I}^{(1)} := diag(\begin{bmatrix} \mathbf{1} \\ \mathbf{0} \end{bmatrix})$, $\mathbf{I}^{(2)} := diag(\begin{bmatrix} \mathbf{0} \\ \mathbf{1} \end{bmatrix})$,

*then the multi-view optimization problem* (4) *can be reformulated in the following convex form*

$$\min_{\substack{\|\mathbf{C}_{:j}\|_2 \leq \gamma_1 \\ \|\mathbf{E}_{:j}\|_2 \leq \gamma_2}} \quad \min_{\mathbf{\Phi}: \begin{bmatrix} \mathbf{C} \\ \mathbf{E} \end{bmatrix} \mathbf{\Phi} = \hat{\mathbf{Z}}} L_1(\mathbf{C}\mathbf{\Phi}; \mathbf{Z}^{(1)}) + L_2(\mathbf{E}\mathbf{\Phi}; \mathbf{Z}^{(2)}) + \lambda \|\mathbf{\Phi}\|_{2,1}$$

$$= \min_{\hat{\mathbf{Z}}} \quad L_1(\hat{\mathbf{Z}}^{(1)}; \mathbf{Z}^{(1)}) + L_2(\hat{\mathbf{Z}}^{(2)}; \mathbf{Z}^{(2)}) + \lambda \max_{0 \leq \eta \leq 1} \|\mathbf{U}_\eta^{-1}\hat{\mathbf{Z}}\|_{tr}$$

*where* $\mathbf{U}_\eta = \frac{\gamma_1}{\sqrt{\eta}}\mathbf{I}^{(1)} + \frac{\gamma_2}{\sqrt{1-\eta}}\mathbf{I}^{(2)}$ *and* $L_i(\mathbf{Y}; \hat{\mathbf{Y}}) = \sum_{t=1}^T \mathcal{L}_i(\mathbf{y}_t; \hat{\mathbf{y}}_t)$. *Moreover, we can show that the regularizer term* $\|\mathbf{U}_\eta^{-1}\hat{\mathbf{Z}}\|_{tr}$ *is concave in* $\eta$. *The trace norm induces a low rank result.*

**Proof:** The proof can be readily derived from the results of White et al. (2012). ∎

In the next section, we demonstrate how to obtain globally optimal estimates of $\mathbf{E}$, $\mathbf{C}$ and $\mathbf{\Phi}$.

**Remark 1:** This maximum likelihood formulation demonstrates how the distributional assumptions on the observations $\mathbf{x}_t$ can be generalized to any exponential family. Once expressed as the above optimization problem, one can further consider other losses and regularizers that may not immediately have a distributional interpretation, but result in improved prediction performance. This generalized formulation of maximum likelihood for LDS, therefore, has the additional benefit that it can flexibly incorporate optimization improvements, such as robust losses.[4] Also a regularizer can be designed to control overfitting to noisy observation, which is an issue in LDS that can result in an unstable latent dynamics estimate (Buesing et al., 2012a). Therefore, by controlling undesired overfitting to noisy samples one can also prevent unintended unstable model identification.

**Remark 2:** We can generalize the optimization further to learn an *LDS with exogenous input*: a control vector $\mathbf{u}_t \in \mathbb{R}^d$ that impacts both the hidden state and observations. This entails adding some new variables to the general LDS model that can be expressed as

$$\phi_{t+1} = \mathbf{A}\phi_t + \mathbf{B}\mathbf{u}_t + \eta_t$$
$$\mathbf{x}_t = \mathbf{C}\phi_t + \mathbf{D}\mathbf{u}_t + \epsilon_t$$

with additional matrices $\mathbf{B} \in \mathbb{R}^{k \times d}$ and $\mathbf{D} \in \mathbb{R}^{d \times d}$. Again by multiplying the state evolution equation by matrix $\mathbf{C}$ the resulting equations are

$$\mathbf{x}_{t+1} = \mathbf{E}\phi_t + \mathbf{F}\mathbf{u}_t + \mathbf{D}\mathbf{u}_{t+1} + \epsilon'_{t+1}$$
$$\mathbf{x}_t = \mathbf{C}\phi_t + \mathbf{D}\mathbf{u}_t + \epsilon_t$$

where $\mathbf{F} := \mathbf{CB}$. Therefore, the loss can be generally expressed as

$$\mathcal{L}_1(\mathbf{E}\phi_{t-1} + \mathbf{F}\mathbf{u}_{t-1} + \mathbf{D}\mathbf{u}_t; \mathbf{x}_t) + \mathcal{L}_2(\mathbf{C}\phi_t + \mathbf{D}\mathbf{u}_t; \mathbf{x}_t).$$

The optimization would now be over the variables $\mathbf{C}, \mathbf{E}, \boldsymbol{\Phi}, \mathbf{D}, \mathbf{F}$, where the optimization could additionally include regularizers on $\mathbf{D}$ and $\mathbf{F}$ to control overfitting. Importantly, the addition of these variables $\mathbf{D}, \mathbf{F}$ does not modify the convexity properties of the loss, and the treatment for estimating $\mathbf{E}, \mathbf{C}$ and $\boldsymbol{\Phi}$ in section 4 directly applies. The optimization problem is jointly convex in $\mathbf{D}, \mathbf{F}$ and any one of $\mathbf{E}, \mathbf{C}$ or $\boldsymbol{\Phi}$ and jointly convex in $\mathbf{D}$ and $\mathbf{F}$. Therefore, an outer minimization over $\mathbf{D}$ and $\mathbf{F}$ can be added to Algorithm 1 and we will still obtain a globally optimal solution.

# 4 LDS Estimation Algorithm

To learn the optimal parameters for the reformulated two-view model, we adopt the generalized conditional gradient (GCG) algorithm developed by Yu et al. (2014). GCG is designed for optimization problems of the form $l(x) + f(x)$ where $l(x)$ is convex and continuously differentiable with Lipschitz continuous gradient and $f(x)$ is a (possibly non-differentiable) convex function. The algorithm is computationally efficient, as well providing a reasonably fast $O(1/t)$ rate of convergence to the global minimizer. Though we have a nonconvex optimization problem, we can use the convex reformulation for two-view low-rank matrix factorization and resulting algorithm in (Yu et al., 2014, Section 4). This algorithm includes a generic local improvement step, which significantly accelerates the convergence of the algorithm to a global optimum in practice. We provide a novel local improvement update, which both speeds learning and enforces a sparser structure on $\boldsymbol{\Phi}$, while maintaining the same theoretical convergence properties of GCG.

In our experiments, we specifically address the setting when the observations are assumed to be Gaussian, giving an $\ell_2$ loss. We also prefer the unconstrained objective function that can be efficiently minimized by fast unconstrained optimization algorithms. Therefore, using the well-established equivalent form of the regularizer (Bach et al., 2008), the objective (4) can be equivalently cast for the Gaussian distributed time series $\mathbf{x}_t$ as

$$\min_{\mathbf{C},\mathbf{E},\boldsymbol{\Phi}} \sum_{t=1}^{T} \|\mathbf{E}\phi_{t-1} - \mathbf{x}_t\|_2^2 + \|\mathbf{C}\phi_t - \mathbf{x}_t\|_2^2 + \lambda \sum_{j=1}^{k} \|\boldsymbol{\Phi}_{j:}\|_2 \ \max(\tfrac{1}{\gamma_1}\|\mathbf{C}_{:j}\|_2, \tfrac{1}{\gamma_2}\|\mathbf{E}_{:j}\|_2). \tag{5}$$

This product form of the regularizer is also preferred over the square form used in (Yu et al., 2014), since it induces row-wise sparsity on $\boldsymbol{\Phi}$. Though the square form $\|\boldsymbol{\Phi}\|_F^2$ admits efficient optimizers due to its smoothness, it does not prefer to zero out rows of $\boldsymbol{\Phi}$ while with the regularizer of the form (5), the learned hidden state will be appropriately projected down to a lower-dimensional space where many dimensions could be dropped from $\boldsymbol{\Phi}$, $\mathbf{C}$ and $\mathbf{E}$ giving a low rank solution. In practice, we found that enforcing this sparsity property on $\boldsymbol{\Phi}$ significantly improved stability.[5] Consequently, we need optimization routines that are appropriate for the non smooth regularizer terms.

The local improvement step involves alternating block coordinate descent between $\mathbf{C}, \mathbf{E}$ and $\boldsymbol{\Phi}$, with an accelerated proximal gradient algorithm (FISTA) (Beck and Teboulle, 2009) for each descent step. To use the FISTA algorithm we need to provide a proximal operator for the non-smooth regularizer in (5).

**Algorithm 1** LDS-DV

---

**Input:** training sequence $\{x_t, t \in [1, T]\}$
**Output:** $\mathbf{C}, \mathbf{A}, \boldsymbol{\phi}_t, \boldsymbol{\Sigma}_\eta, \boldsymbol{\Sigma}_\epsilon$
Initialize $\mathbf{C}_0, \mathbf{E}_0, \boldsymbol{\Phi}_0$
$\mathbf{U}_1 \leftarrow [\mathbf{C}_0^\top; \mathbf{E}_0^\top]^\top, \quad \mathbf{V}_1 \leftarrow \boldsymbol{\Phi}_0^\top$
**for** $i = 1, \ldots$ **do**
    $(\mathbf{u}_i, \mathbf{v}_i) \leftarrow \arg\min_{\mathbf{uv}^\top \in \mathcal{A}} \langle \nabla\ell(\mathbf{U}_i, \mathbf{V}_i), \mathbf{uv}^\top \rangle$ // compute polar
    $(\eta_i, \theta_i) \leftarrow \arg\min_{0 \le \eta \le 1, \theta \ge 0} \ell((1-\eta)\mathbf{U}_i\mathbf{V}_i^\top + \theta\mathbf{u}_i\mathbf{v}_i^\top) + \lambda((1-\eta)\rho_i + \theta)$ // partially corrective up-
    date (PCU)
    $\mathbf{U}_{init} \leftarrow [\sqrt{1-\eta_i}\mathbf{U}_i, \sqrt{\theta_i}\mathbf{u}_i], \mathbf{V}_{init} \leftarrow [\sqrt{1-\eta_i}\mathbf{V}_i, \sqrt{\theta_i}\mathbf{v}_i]$
    $(\mathbf{U}_{i+1}, \mathbf{V}_{i+1}) \leftarrow \text{FISTA}(\mathbf{U}_{init}\mathbf{V}_{init})$
    $\rho_i = \frac{1}{2}\sum_{j=1}^{i+1}(\|(\mathbf{U}_{i+1})_{:i}\|_{2v}^2 + \|(\mathbf{V}_{i+1})_{:i}\|_2^2)$
**end for**
$(\mathbf{C}; \mathbf{E}) \leftarrow \mathbf{U}_{i+1}, \quad \boldsymbol{\Phi} \leftarrow \mathbf{V}_{i+1}^\top$
$\mathbf{A} \leftarrow \boldsymbol{\Phi}_{2:T} * \boldsymbol{\Phi}_{1:T-1}^\dagger$
estimate $\boldsymbol{\Sigma}_\eta, \boldsymbol{\Sigma}_\epsilon$ by sample covariances

---

Let the proximal operator of a convex and possibly non-differentiable function $\lambda f(\mathbf{y})$ be defined as

$$\text{prox}_{\lambda f}(\mathbf{x}) = \arg\min_{\mathbf{y}} \lambda f(\mathbf{y}) + \tfrac{1}{2}\|\mathbf{x} - \mathbf{y}\|_2^2.$$

FISTA is an accelerated version of ISTA (Iterative Shrinkage-Thresholding Algorithm) that it-eratively performs a gradient descent update with the smooth component of the objective, and then applies the proximal operator as a projection step. Each iteration updates the variable $\mathbf{x}$ as $\mathbf{x}^{k+1} = \text{prox}_{\lambda_k f}(\mathbf{x}^k - \lambda_k \nabla l(\mathbf{x}^k))$, which converges to a fixed point. If there is no known form for the proximal operator, as is the case for our non-differentiable regularizer, a common strategy is to numerically calculate the proximal update. This approach, however, can be prohibitively expensive, and an analytic (closed) form is clearly preferable. We derive such a closed form for (5) in Theorem 1.

**Theorem 1** *For a vector* $\mathbf{v} = \begin{bmatrix} \mathbf{v}_1 \\ \mathbf{v}_2 \end{bmatrix}$ *composed of two subvectors* $\mathbf{v}_1, \mathbf{v}_2$, *define* $f(\mathbf{v}) = \lambda\|\mathbf{v}\|_{2v} := \lambda\max(\|\mathbf{v}_1\|_2, \|\mathbf{v}_2\|_2)$. *The proximal operator for this function is*

$$prox_f(\mathbf{v}) = \begin{cases} \begin{bmatrix} \mathbf{v}_1 \max\{1 - \frac{\alpha}{\|\mathbf{v}_1\|}, 0\} \\ \mathbf{v}_2 \max\{1 - \frac{\lambda-\alpha}{\|\mathbf{v}_2\|}, 0\} \end{bmatrix} & \text{if } \|\mathbf{v}_1\| \le \|\mathbf{v}_2\| \\ \begin{bmatrix} \mathbf{v}_1 \max\{1 - \frac{\lambda-\beta}{\|\mathbf{v}_1\|}, 0\} \\ \mathbf{v}_2 \max\{1 - \frac{\beta}{\|\mathbf{v}_2\|}, 0\} \end{bmatrix} & \text{if } \|\mathbf{v}_2\| \le \|\mathbf{v}_1\| \end{cases}$$

*where* $\alpha := \max\{.5(\|\mathbf{v}_1\| - \|\mathbf{v}_2\| + \lambda), 0\}$ *and* $\beta := \max\{.5(\|\mathbf{v}_2\| - \|\mathbf{v}_1\| + \lambda), 0\}$.

**Proof:** See Appendix A. ∎

This result can be further generalized to enable additional regularization components on $\mathbf{C}$ and $\mathbf{E}$, such as including an $\ell_1$ norm on each column to further enforce sparsity (such as in the elastic net). There is no closed form for the proximal operator of the sum of two functions in general. We prove, however, that for special case of a linear combination of the two-view norm with any norms on the columns of $\mathbf{C}$ and $\mathbf{E}$, the proximal mapping reduces to a simple composition rule.

**Theorem 2** *For norms* $R_1(\mathbf{v}_1)$ *and* $R_2(\mathbf{v}_2)$, *the proximal operator of the linear combination* $R_c(\mathbf{v}) = \lambda\|\mathbf{v}\|_{2v} + \nu_1 R_1(\mathbf{v}_1) + \nu_2 R_2(\mathbf{v}_2)$ *for* $\nu_1, \nu_2 \ge 0$ *admits the simple composition* $prox_{R_c}(\mathbf{v}) = prox_{\lambda\|.\|_{2v}} \left( \begin{bmatrix} prox_{\nu_1 R_1}(\mathbf{v}_1) \\ prox_{\nu_2 R_2}(\mathbf{v}_2) \end{bmatrix} \right)$.

**Proof:** See Appendix A. ∎

### 4.1 Recovery of the LDS model parameters

The above reformulation provides a tractable learning approach to obtain the optimal parameters for the two-view reformulation of LDS; given this optimal solution, we can then estimate the parameters

to the original LDS. The first step is to estimate the transition matrix $\mathbf{A}$. A natural approach is to use (2), and set $\hat{\mathbf{A}} = \hat{\mathbf{C}}^\dagger \hat{\mathbf{E}}$ for pseudoinverse $\hat{\mathbf{C}}^\dagger$. This $\hat{\mathbf{A}}$, however, might be sensitive to inaccurate estimation of the (effective) hidden state dimension $k$. We found in practice that modifications from the optimal choice of $k$ might result in unstable solutions and produce unreliable forecasts. Instead, a more stable $\hat{\mathbf{A}}$ can be learned from the hidden states themselves. This approach also focuses estimation of $\mathbf{A}$ on the forecasting task, which is our ultimate aim.

Given the sequence of hidden states, $\phi_1, \ldots, \phi_T$, there are several strategies that could be used to estimate $\mathbf{A}$, including simple autoregressive models to more sophisticated strategies (Siddiqi et al., 2007). We opt for a simple linear regression solution $\hat{\mathbf{A}} = \arg\min_{\mathbf{A}} \sum_{t=1}^{T-1} \|\phi_{t+1} - \mathbf{A}\phi_t\|_2^2$ which we found produced stable $\hat{\mathbf{A}}$.

To estimate the noise parameters $\mathbf{\Sigma}_\eta, \mathbf{\Sigma}_\epsilon$, recall $\eta_t = \phi_{t+1} - \hat{\mathbf{A}}\phi_t$, $\epsilon_t = \mathbf{x}_t - \mathbf{C}\phi_t$. Having obtained $\hat{\mathbf{A}}$, therefore, we can estimate the noise covariance matrices by computing their sample covariances as $\hat{\mathbf{\Sigma}}_\eta = \frac{1}{T-1}\sum_{t=1}^{T}\eta_t\eta_t^\top$, $\hat{\mathbf{\Sigma}}_\epsilon = \frac{1}{T-1}\sum_{t=1}^{T}\epsilon_t\epsilon_t^\top$. The final LDS learning procedure is outlined in Algorithm 1. For more details about polar computation and partially corrective subroutine see (Yu et al., 2014, Section 4).

## 5    Experimental results

We evaluate the proposed algorithm by comparing one step prediction performance and computation speed with alternative methods for real and synthetic time series. We report the normalized mean square error (NMSE) defined as NMSE $= \frac{\sum_{t=1}^{T_{test}} \|y_t - \hat{y}_t\|^2}{\sum_{t=1}^{T_{test}} \|y_t - \mu_y\|^2}$ where $\mu_y = \frac{1}{T_{test}}\sum_{t=1}^{T_{test}} y_t$.

**Algorithms:** We compared the proposed algorithm to a well-established method-of moment-based algorithm, N4SID (Van Overschee and De Moor, 1994), Hilbert space embeddings of hidden Markov models (HSE-HMM) (Song et al., 2010), expectation-maximization for estimating the parameters of a Kalman filter (EM) (Roweis and Ghahramani, 1999) and PEM (Ljung, 1999). These are standard baseline algorithms that are used regularly for LDS identification. The estimated parameters by N4SID were used as the initialization point for EM and PEM algorithms in our experiments. We used the built-in functions, n4sid and pem, in Matlab, with the order selected by the function, for the subspace identification method and PEM, respectively. For our algorithm, we select the regularization parameter $\lambda$ using cross-validation. For the time series, the training data is split by performing the learning on first $80\%$ of the training data and evaluating the prediction performance on the remaining $20\%$.

**Real datasets:** For experiments on real datasets we select the climate time series from IRI data library that recorded the surface temperature on the monthly basis for tropical Atlantic ocean (ATL) and tropical Pacific ocean (CAC). In CAC we selected first $30 \times 30$ grids out of the total $84 \times 30$ locations with 399 monthly samples, while in ATL the first $9 \times 9$ grids out of the total $38 \times 25$ locations are selected each with timeseries of length $564$. We partitioned each area to smaller areas of size $3 \times 3$ and arrange them to vectors of size 9, then seasonality component of the time series are removed and data is centered to have zero mean. We ran two experiments for each dataset. For the first, the whole sequence is sliced into $70\%$ training and $30\%$ test. For the second, a short training set of 70 samples is selected, with a test sequence of size $50$.

**Synthetic datasets:** In the synthetic experiments, the datasets are generated by an LDS model (1) of different system orders, $k$, and observation sizes, $d$. For each test case, 100 data sequences of length 200 samples are generated and sliced to $70\%$, $30\%$ ratios for training set and test set, respectively. The dynamics matrix $\mathbf{A}$ is selected to produce a stable system: $\{|\sigma_i(\mathbf{A})| = s : s \leq 1, \forall i \in (1,k)\}$ where $\sigma_i(\mathbf{A})$ is the $i$th eigen value of matrix $\mathbf{A}$. The noise components are drawn from Gaussian distributions and scaled so that $p_\eta := E\{\eta^\top\eta\}/m$ and $p_\epsilon := E\{\epsilon^\top\epsilon\}/n$. Each test is repeated with the following settings: {**S1:** $s = 0.970, p_\eta = 0.50$ and $p_\epsilon = 0.1$}, {**S2:** $s = 0.999, p_\eta = 0.01$ and $p_\epsilon = 0.1$}.

**Results:** The NMSE and run-time results obtained on real and synthetic datasets are shown in Table 1 and Table 2, respectively. In terms of NMSE, LDS-DV outperforms and matches the alternative methods. In terms of algorithm speed, the LDS-DV learns the model much faster than the competitors and scales well to larger dimension models. The speed improvement is more significant for larger datasets and observations with higher dimensions.

Table 1: Real time series

| | ATL(Long) | | ATL(Short) | | CAC(Long) | | CAC(Short) | |
|---|---|---|---|---|---|---|---|---|
| | NMSE | Time | NMSE | Time | NMSE | Time | NMSE | Time |
| **LDS-MV** | **0.45**$_{\pm0.03}$ | 0.26 | **0.54**$_{\pm0.05}$ | 0.22 | **0.58**$_{\pm0.02}$ | 0.28 | **0.63**$_{\pm0.03}$ | 0.14 |
| **N4SID** | **0.52**$_{\pm0.04}$ | 2.34 | **0.59**$_{\pm0.05}$ | 0.95 | **0.61**$_{\pm0.02}$ | 1.23 | 0.84$_{\pm0.07}$ | 1.08 |
| **EM** | 0.64$_{\pm0.04}$ | 7.87 | 0.88$_{\pm0.07}$ | 3.92 | 0.81$_{\pm0.02}$ | 5.70 | 1.02$_{\pm0.08}$ | 4.12 |
| **HSE-HMM** | 675.87$_{\pm629.46}$ | 0.79 | 0.97$_{\pm0.01}$ | 0.16 | 11.24$_{\pm8.23}$ | 0.39 | 2.82$_{\pm1.60}$ | 0.17 |
| **PEM-SSID** | 0.71$_{\pm0.08}$ | 20.00 | 1.52$_{\pm0.66}$ | 16.38 | 1.38$_{\pm0.15}$ | 19.67 | 2.68$_{\pm0.78}$ | 20.58 |

Table 2: Synthetic time series

| | (S1) d=5 , k=3 | | (S2) d=5 , k=3 | | (S1) d=8 , k=6 | | (S2) d=8 , k=6 | | (S1) d=16 , k=9 | | (S2) d=16 , k=9 | |
|---|---|---|---|---|---|---|---|---|---|---|---|---|
| | NMSE | Time | NMSE | Time | NMSE | Time | NMSE | Time | NMSE | Time | NMSE | Time |
| **LDS-MV** | **0.12**$_{\pm0.01}$ | 0.49 | **0.17**$_{\pm0.02}$ | 0.36 | **0.08**$_{\pm0.00}$ | 0.66 | **0.04**$_{\pm0.00}$ | 0.52 | **0.07**$_{\pm0.00}$ | 1.01 | **0.03**$_{\pm0.00}$ | 1.72 |
| **N4SID** | **0.12**$_{\pm0.01}$ | 0.81 | 0.42$_{\pm0.04}$ | 0.76 | 0.11$_{\pm0.00}$ | 1.45 | 0.39$_{\pm0.04}$ | 1.38 | 0.10$_{\pm0.00}$ | 4.29 | 0.42$_{\pm0.04}$ | 4.40 |
| **EM** | 0.18$_{\pm0.01}$ | 4.99 | **0.15**$_{\pm0.02}$ | 4.62 | 0.14$_{\pm0.01}$ | 6.01 | **0.04**$_{\pm0.00}$ | 5.03 | 0.13$_{\pm0.00}$ | 19.21 | 0.03$_{\pm0.00}$ | 19.83 |
| **HSE-HMM** | 2.4e+4$_{\pm1.7e+4}$ | 0.48 | 2.2e+7$_{\pm2.2e+7}$ | 0.50 | 7.8e+03$_{\pm7.7e+03}$ | 0.49 | 0.65$_{\pm0.02}$ | 0.55 | 22.92$_{\pm21.83}$ | 0.53 | 0.71$_{\pm0.01}$ | 0.61 |
| **PEM-SSID** | 0.14$_{\pm0.01}$ | 10.72 | 0.25$_{\pm0.03}$ | 9.08 | 0.12$_{\pm0.01}$ | 15.22 | 0.08$_{\pm0.01}$ | 13.97 | 0.09$_{\pm0.01}$ | 38.39 | 0.06$_{\pm0.02}$ | 41.10 |

Results for real and synthetic datasets are listed in Table 1 and Table 2, respectively. The first column of each dataset is the average normalized MSE with standard error and the second column is the algorithm runtime in CPU seconds. The best NMSE according to pairwise $t$-test with significance level of 5% is highlighted.

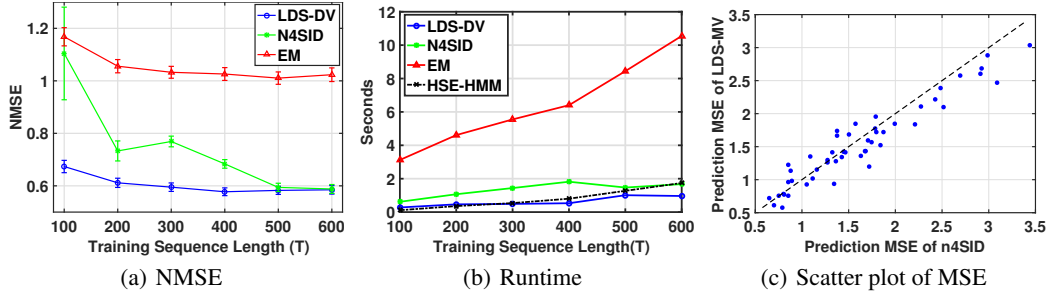

(a) NMSE      (b) Runtime      (c) Scatter plot of MSE

Figure 2: a) NMSE of the LDS-DV for increasing length of training sequence. The difference between LDS-DV and N4SID is more significant in shorter training length, while both converge to the same accuracy in large $T$. HSE-HMM is omitted due to its high error. b) Runtime in CPU seconds for increasing length of training sequence. LDS-DV scales well with large sample length. c) MSE of the LDS-DV versus MSE of N4SID. In higher values of MSE, the points are below identity function line and LDS-DV is more likely to win.

For test cases with $|\sigma_i(\mathbf{A})| \simeq 1$, designed to evaluate the prediction performance of the methods for marginally stable systems, LDS-DV still can learn a stable model while the other algorithms might not learn a stable model. The proposed LDS-DV method does not explicitly impose stability, but the regularization favors $\mathbf{A}$ that is stable. The regularizer on latent state encourages smooth dynamics and controls overfitting: overfitting to noisy observations can lead to unstable estimate of the model (Buesing et al., 2012a), and a smooth latent trajectory is a favorable property in most real-world applications.

Figure 2(c) shows the MSE of LDS-DV versus N4SID, for all the CAC time-series. This figure illustrates that for easier problems, LDS-DV and N4SID are more comparable. However, as the difficulty increase, and MSE increases, LDS-DV begins to consistently outperform N4SID.

Figures 2(a) and 2(b) illustrate the accuracy and runtime respectively of the algorithms versus training length. We used the synthetic LDS model under condition S1 with $n = 8$, $m = 6$. Values are averaged over 20 runs with a test length of 50 samples. LDS-DV has better early performance, for smaller sample sizes. At larger sample sizes, they reach approximately the same error level.

## 6 Conclusion

In this paper, we provided an algorithm for optimal estimation of the parameters for a time-invariant, discrete-time linear dynamical system. More precisely, we provided a reformulation of the model as a two-view objective, which allowed recent advances for optimal estimation for two-view models to be applied. The resulting algorithm is simple to use and flexibly allows different losses and regularizers

to be incorporated. Despite this simplicity, significant improvements were observed over a widely accepted method for subspace identification (N4SID), both in terms of accuracy for forecasting and runtime.

The focus in this work was on forecasting, therefore on optimal estimation of the hidden states and transition matrices; however, in some settings, estimation of noise parameters for LDS models is also desired. An unresolved issue is joint optimal estimation of these noise parameters. Though we do explicitly estimate the noise parameters, we do so only from the residuals after obtaining the optimal hidden states and transition and observation matrices. Moreover, consistency of the learned parameters by the proposed procedure of this paper is still an open problem and will be an interesting future work.

The proposed optimization approach for LDSs should be useful for applications where alternative noise assumptions are desired. A Laplace assumption on the observations, for example, provides a more robust $\ell_1$ loss. A Poisson distribution has been advocated for count data, such as for neural activity, where the time series is a vector of small integers (Buesing et al., 2012b). The proposed formulation of estimation for LDSs easily enables extension to such distributions. An important next step is to investigate the applicability to a wider range of time series data.

### Acknowledgments

This work was supported in part by the Alberta Machine Intelligence Institute and NSERC. During this work, M. White was with the Department of Computer Science, Indiana University.

## Footnotes

[1] The balanced averaging can be generalized to a convex combination of the log-likelihood which adds a flexibility to the problem that can be tuned to improve performance. However, we found that the simple balanced combination renders the best experimental performance in most cases.

[2] Consult Banerjee et al. (2005) for a complete overview of this correspondence.

[3] Throughout this paper, $\mathbf{X}_{i:}$ ($\mathbf{X}_{:i}$) is used to denote the *i*th row (*i*th column) of matrix $\mathbf{X}$ and also $[\mathbf{X}; \mathbf{Y}]$ ([$\mathbf{x}; \mathbf{y}$]) denotes the matrix (vector) concatenation operator which is equal to $[\mathbf{X}^\top, \mathbf{Y}^\top]^\top$ ($[\mathbf{x}^\top, \mathbf{y}^\top]^\top$).

[4] Thus, we used $\mathcal{L}_1$ and $\mathcal{L}_2$ in (4) to generally refer to any loss function that is convex in its first argument.

[5]This was likely due to a reduction in the size of the transition parameters, resulting in improved re-estimation of $\mathbf{A}$ and a corresponding reduction in error accumulation when using the model for forecasting.

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
