[Supplementary Material]

# A    Proof of Theorems 1 and 2

**Proof: [Theorem 1]** Let $x = [x_1; x_2]$ $y = [\mathbf{v}_1; \mathbf{v}_2]$ then the proximal operator of $f(x) = \lambda \max(\|x_1\|_2, \|x_2\|_2)$ can be expressed as

$$u = \text{prox}_{f(\mathbf{v})} = \arg\min_x \frac{1}{2}\|x - y\|_2^2 + f(x)$$

$$= \arg\min_{x_1, x_2} \frac{1}{2}\|x_1 - \mathbf{v}_1\|_2^2 + \frac{1}{2}\|x_2 - \mathbf{v}_2\|_2^2$$

$$+ \lambda \max(\|x_1\|_2, \|x_2\|_2). \tag{6}$$

Now, splitting the search space into two subspace $S_1 = \{x | \|x_1\|_2 \geq \|x_2\|_2\}$ and its complement space $S_2 = \{x | \|x_2\|_2 \geq \|x_1\|_2\}$. If we confine our search to subspace $S_1$ the optimization problem (6) can be reformulated as the following convex constraint problem

$$\min_{x_1, x_2} \frac{1}{2}\|x_1 - \mathbf{v}_1\|_2^2 + \frac{1}{2}\|x_2 - \mathbf{v}_2\|_2^2 + \lambda\|x_1\|_2.$$

$$\text{subject to } \|x_1\|_2 \geq \|x_2\|_2 \tag{7}$$

According to KKT optimality conditions for convex problem (Boyd and Vandenberghe, 2004), a point $x^* = [x_1^*; x_2^*]$ is optimal point of this optimization problem if the following conditions are satisfied:

$$(x_2^* - \mathbf{v}_2) + \nu x_2^* / \|x_2^*\|_2 = 0$$
$$(x_1^* - \mathbf{v}_1) + \lambda x_1^* / \|x_1^*\|_2 - \nu x_1^* / \|x_1^*\|_2 = 0$$
$$\|x_1^*\|_2 - \|x_2^*\|_2 \geq 0$$
$$\nu(\|x_1^*\|_2 - \|x_2^*\|_2) = 0$$
$$\nu \geq 0 \tag{8}$$

where $\nu$ is the Lagrange multiplier for the inequality constraint. Solving (8) for $x_1^*$, $x_2^*$ and $\nu$ one can readily obtain

$$\begin{bmatrix} x_1^* = \mathbf{v}_1 * \max\{1 - \frac{\nu}{\|\mathbf{v}_1\|}, 0\} \\ x_2^* = \mathbf{v}_2 * \max\{1 - \frac{\lambda - \nu}{\|\mathbf{v}_2\|}, 0\} \end{bmatrix} \tag{9}$$

According to the slackness condition () if $\|x_1^*\|_2 - \|x_2^*\|_2 \geq 0$ then $\nu = 0$ or if $\nu > 0$ then $\|x_1^*\|_2 = \|x_2^*\|_2$. Therefore the optimal $\nu$ can be obtained as

$$\begin{cases} \nu = 0 & \text{if } \|\mathbf{v}_1\| + \lambda < \|\mathbf{v}_2\| \\ \nu = .5(\|\mathbf{v}_1\| - \|\mathbf{v}_2\| + \lambda) & \text{if } \|\mathbf{v}_1\| \leq \|\mathbf{v}_2\| \leq \|\mathbf{v}_1\| + \lambda \end{cases}$$

Hence, the optimum solution under $S_1$ is

$$\begin{cases} \begin{bmatrix} x_1^* = \mathbf{v}_1 \\ x_2^* = \mathbf{v}_2 * \max\{1 - \frac{\lambda}{\|\mathbf{v}_2\|}, 0\} \end{bmatrix} & \text{if } \|\mathbf{v}_1\| + \lambda < \|\mathbf{v}_2\| \\ \begin{bmatrix} x_1^* = \mathbf{v}_1 * \max\{1 - \frac{\nu}{\|\mathbf{v}_1\|}, 0\} \\ x_2^* = \mathbf{v}_2 * \max\{1 - \frac{\lambda - \nu}{\|\mathbf{v}_2\|}, 0\} \end{bmatrix} & \text{if } \|\mathbf{v}_1\| \leq \|\mathbf{v}_2\| \leq \|\mathbf{v}_1\| + \lambda \end{cases} \tag{10}$$

We can repeat the same approach to obtain the optimal solution for the complement subspace $S_2$. ∎

**Lemma 2** *Let $G_{\lambda f}(x, v) := \frac{1}{2}\|x - v\|_2^2 + \lambda f(x)$ therefore the Moreau envelope of function $\lambda f$ is defined as $\text{M}_{\lambda f}(\mathbf{v}) := \min_x G_{\lambda f}(x, v)$ (Parikh and Boyd, 2013).*
*a) $\text{M}_{\lambda f}(\mathbf{v}) = G_{\lambda f}(\text{prox}_{\lambda f(\mathbf{v})}, v)$*
*b) If $f(x) = \|x\|_{2v} = \max(\|x_1\|_2, \|x_2\|_2)$ we have*

$$\text{M}_{\lambda\|.\|_{2v}}(\mathbf{v}) = \max_{0 \leq \gamma \leq 1} \text{M}_{\lambda\gamma\|.\|_2}(\mathbf{v}_1) + \text{M}_{\lambda(1-\gamma)\|.\|_2}(\mathbf{v}_2) \tag{11}$$

**Proof:** a) This is simply follows from the definition of proximal operator.
b) We can simply show that

$$\|x\|_{2v} = \max_{0 \leq \gamma \leq 1} (\gamma\|x_1\|_2 + (1 - \gamma)\|x_2\|_2)$$

then its Moreau envelop is

$$\mathrm{M}_{R_c(\mathbf{v})}(\mathbf{v}) = \min_{x_1,x_2} \frac{1}{2}\|x_1 - \mathbf{v}_1\|_2^2 + \frac{1}{2}\|x_2 - \mathbf{v}_2\|_2^2$$
$$+ \lambda \max_{0 \leq \gamma \leq 1} (\gamma\|x_1\|_2 + (1-\gamma)\|x_2\|_2)$$
$$= \max_{0 \leq \gamma \leq 1} \min_{x_1} \frac{1}{2}\|x_1 - \mathbf{v}_1\|_2^2 + \lambda\gamma\|x_1\|_2$$
$$+ \min_{x_2} \frac{1}{2}\|x_2 - \mathbf{v}_2\|_2^2 + (1-\gamma)\lambda\|x_2\|_2$$
$$= \max_{0 \leq \gamma \leq 1} \mathrm{M}_{\lambda\gamma\|.\|_2}(\mathbf{v}_1) + \mathrm{M}_{\lambda(1-\gamma)\|.\|_2}(\mathbf{v}_2)$$

∎

**Proof:** [**Theorem 2**] The Moreau envelope of $R_c(\mathbf{v})$ is

$$\mathrm{M}_{R_c(\mathbf{v})}(\mathbf{v}) = \min_{x_1,x_2} \frac{1}{2}\|x_1 - \mathbf{v}_1\|_2^2 + \frac{1}{2}\|x_2 - \mathbf{v}_2\|_2^2$$
$$+ \lambda \max_{0 \leq \gamma \leq 1} (\gamma\|x_1\|_2 + (1-\gamma)\|x_2\|_2) + \nu_1 R_1(\mathbf{v}_1) + \nu_2 R_2(\mathbf{v}_2)$$
$$= \max_{0 \leq \gamma \leq 1} \min_{x_1} \frac{1}{2}\|x_1 - \mathbf{v}_1\|_2^2 + \lambda\gamma\|x_1\|_2 + \nu_1 R_1(\mathbf{v}_1)$$
$$+ \min_{x_2} \frac{1}{2}\|x_2 - \mathbf{v}_2\|_2^2 + (1-\gamma)\lambda\|x_2\|_2 + \nu_2 R_2(\mathbf{v}_2)$$
$$= \max_{0 \leq \gamma \leq 1} \mathrm{M}_{\lambda\gamma\|.\|_2 + \nu_1 R_1}(\mathbf{v}_1) + \mathrm{M}_{\lambda(1-\gamma)\|.\|_2 + \nu_2 R_2}(\mathbf{v}_2)$$

(12)

Let $h_1(\mathbf{v}_1) := \lambda\gamma\|\mathbf{v}_1\|_2 + \nu_1 R_1(\mathbf{v}_1)$ and $h_2(\mathbf{v}) := \lambda(1-\gamma)\|\mathbf{v}_2\|_2 + \nu_2 R_2(\mathbf{v}_2)$. From (Haeffele et al., 2014, Theorem 3), we know that $\mathrm{prox}_{h_1}(\mathbf{v}_1) = \mathrm{prox}_{\lambda\gamma\|.\|_2}(\mathrm{prox}_{\nu_1 R_1}(\mathbf{v}_1))$ and $\mathrm{prox}_{h_2}(\mathbf{v}_2) = \mathrm{prox}_{\lambda(1-\gamma)\|.\|_2}(\mathrm{prox}_{\nu_2 R_2}(\mathbf{v}_2))$, and so

$$\mathrm{M}_{h_1}(\mathbf{v}_1) = \mathrm{M}_{\lambda\gamma\|.\|_2}(\mathrm{prox}_{\nu_1 R_1}(\mathbf{v}_1))$$
$$\mathrm{M}_{h_2}(\mathbf{v}_2) = \mathrm{M}_{\lambda(1-\gamma)\|.\|_2}(\mathrm{prox}_{\nu_2 R_2}(\mathbf{v}_2))$$

(13)

Then based on (12) and (13), we obtain

$$\mathrm{M}_{R_c(\mathbf{v})}(\mathbf{v}) = \max_{0 \leq \gamma \leq 1} \mathrm{M}_{\lambda\gamma\|.\|_2}(\mathrm{prox}_{\nu_1 R_1}(\mathbf{v}_1))$$
$$+ \mathrm{M}_{\lambda(1-\gamma)\|.\|_2}(\mathrm{prox}_{\nu_2 R_2}(\mathbf{v}_2))$$

(14)

Finally, based on above equation and (11), we conclude the following composition rule for

$$\mathrm{M}_{R_c(\mathbf{v})}(\mathbf{v}) = \mathrm{M}_{\lambda\|.\|_{2v}}([\mathrm{prox}_{\nu_1 R_1}(\mathbf{v}_1); \ \mathrm{prox}_{\nu_2 R_2}(\mathbf{v}_2)])$$

and according to Lemma 1 the proximal operator is

$$\mathrm{prox}_{R_c}(\mathbf{v}) = \mathrm{prox}_{\lambda\|.\|_{2v}}([\mathrm{prox}_{\nu_1 R_1}(\mathbf{v}_1); \ \mathrm{prox}_{\nu_2 R_2}(\mathbf{v}_2)]).$$

∎

Table 3: Synthetic time series

| | Bernoulli d=5 , k=3 | | Bernoulli d=8 , k=6 | | Bernoulli d=16 , k=9 | | Poisson d=5 ,k=3 | | Poisson d=8 , k=6 | |
|---|---|---|---|---|---|---|---|---|---|---|
| | $GOF_b$ | Time | $GOF_b$ | Time | $GOF_b$ | Time | $GOF_p$ | Time | $GOF_p$ | Time |
| **LDS-MV** | **0.66**±**0.01** | 3.03 | **0.59**±**0.01** | 2.40 | **0.51**±**0.01** | 2.52 | **0.59**±**0.02** | 0.61 | **0.51**±**0.02** | 1.70 |
| **N4SID** | 0.77±0.01 | 1.01 | 0.82±0.01 | 1.68 | 0.75±0.01 | 4.63 | 1.40±0.21 | 0.36 | 1.59±0.33 | 0.49 |
| **EM** | 0.75±0.01 | 2.46 | 0.67±0.01 | 4.61 | 0.63±0.01 | 24.17 | 1.60±0.34 | 1.83 | 2.53±0.60 | 2.48 |

The first column of each dataset is the average goodness-of-fit (GOF) for one step prediction with standard error and the second column is the algorithm runtime in CPU seconds. The best GOF according to pairwise *t*-test with significance level of 5% is highlighted.

# B   Experimental results for discrete value time series

One of the major advantages of formulation (4) is its natural flexibility to encompass any convex loss function such as the Bregman divergences that associate with exponential family distributions and can express a broad range of data property with non-linear transfers. An application that gains benefits from the aforementioned property is to model the count data process with generalized LDS model and consequently adopting the two view formulation to identify the model parameters. An integer-valued stochastic process, that explains the number of occurrence of one phenomenon, can be properly modeled by Poisson distribution (Macke et al., 2015). Therefore, the LDS with Poisson distributed observation can be expressed as:

$$\phi_{t+1} = \mathbf{A}\phi_t + \eta_t$$
$$\mathbf{z}_t = \mathbf{f}(\mathbf{C}\phi_t)$$
$$\mathbb{P}(\mathbf{x}_{i,t}|\mathbf{z}_{i,t}) = \frac{1}{\mathbf{x}_{i,t}!}(\mathbf{z}_{i,t})^{\mathbf{x}_{i,t}} \exp(-\mathbf{x}_{i,t}) \tag{15}$$

where $\mathbf{f}(\boldsymbol{\theta}) = \exp(\boldsymbol{\theta})$. The exponential mapping is not only a natural choice in applications such as neural spike-rate modeling, as explained in (Macke et al., 2015), it also matches with the transfer function associated with the Poisson distribution. Therefore, the negative log-likelihood loss for this model can be characterized by the Bregman divergence, defined as $D_F(\hat{\mathbf{z}}\|\mathbf{z}) := F(\hat{\mathbf{z}}) - F(\mathbf{z}) - \mathbf{f}(\mathbf{z})^\top(\hat{\mathbf{z}} - \mathbf{z})$ where $F(\boldsymbol{\theta}) = \mathbf{1}^\top \exp(\boldsymbol{\theta})$ ($\mathbf{f}(\boldsymbol{\theta}) = \exp(\boldsymbol{\theta})$) is potential (transfer) function corresponding to Poisson distribution.

In Table 3, we compare the performance of the LDS-DV method against the standard N4SID and EM for synthetic time series setting.

For boolean setting, data are sampled from Bernoulli distribution whose mean is changed according to non-linear transfer function of the LDS model where sigmoid transfer function $\mathbf{f}(\boldsymbol{\theta}) = (1 + \exp(-\boldsymbol{\theta}))^{-1}$ is used (Banerjee et al., 2005). Each test case is averaged over 100 data sequences where data are generated similar to synthetic setting **S1** of section 5. For Poisson setting, data are sampled based on model (15) where the final results averaged over 30 data sequences.

Goodness-of-fit for the Bernoulli distribution is the misclassification error: $\text{GOF}_b = \frac{1}{Td}\sum_{t=1}^{T_{test}}\|y_t \neq g(\hat{\mathbf{z}}_t)\|_1$, $g(\theta) = \mathbf{I}_{\theta \geq 0.5}$. And for the Poisson distribution, we define goodness-of-fit as $\text{GOF}_p = \frac{1}{Td}\sum_{t=1}^{T_{test}}\|y_t - h(\hat{\mathbf{z}}_t)\|_1$, $h(\hat{z}) = \text{mode}(P(\hat{z})) = \max_x p(x|\mu = \hat{z})$.

This results are just some primitive results to show the capability of the proposed method in modeling generalized-LDS models.