[Reviews · NeurIPS 2017]

Reviewer 1



The authors introduce a novel method to do MAP estimation of an linear dynamical system. This model is derived through a reformulation of maximum likelihood estimation which introduces two possible parametrizations of the LDS. Experimentally, this methods provides better parameter estimates and predictions than previous methods, and also runs faster on average. Clarity: This paper is clear, but suffers somewhat from its condensed format. In particular, Algorithm 1 should ideally be provided with a larger font size Quality: This paper is of good quality. Originality: This paper introduces a novel approach to estimating the parameters of a linear dynamical system using MLE; previous methods based on EM suffer from poor local minima. Significance: This paper seems somewhat significant. Detailed comments: - Some mathematical results would be clearer with a line or two more that explain the computations, e.g. lines 106-107. - Line 102: would there be cases where it might be valuable to consider another convex combination of the two log-likelihoods? - Instead of using the L2,1 norm to induce sparsity, how would inducing a direct rank constraint by considering matrices of the form VV^T with V rectangular of rank k < n impact your approach? - Lemma 1: could you please define Z and not only \tilde Z? It also seems that at the second line of the equation, you have inverted L_2 and L_1 compared to Eq. (4), is this correct? - What is the complexity of Algorithm 1? Minor comments: - line 100: "that ignoreS" - line 159: "a nonconvex optimization problem" - Please increase the font size for Algorithm 1 and for the legend/axis labels in Figure 2. - Please fix the citations from arXiv so that they have the same format; in particular, lines 336 and 339.

Reviewer 2



This paper proposes an efficient maximum likelihood algorithm for parameter estimation in linear dynamical systems. The problem is reformulated as a two-view generative model with a shared latent factor, and approximated as a matrix factorization problem. The paper then proposes a novel proximal update. Experiments validate the effectiveness of the proposed method. The paper realizes that maximum likelihood style algorithms have some merit over classical moment-matching algorithms in LDS, and wants to solve the efficiency problem of existing maximum likelihood algorithms. Then the paper proposes a theoretical guaranteed proximal update to solve the optimization problem. However, I do not understand why the paper tell the story from a two-view aspect. In LDS, we can construct x_t from two ways. One is from \phi_t and the other is from \phi_{t-1}. Eq.4 is minimizing the reconstruction error for x_t constructed in both ways, with regularization on \Phi, C and A. This is a general framework, which is a reconstruction loss plus regularization, widely used in many classical machine learning algorithms. I do not see much novelty in the proposed learning objective Eq.4, and I cannot tell the merit of stating the old story from a two-view direction. Lemma 1 proposes a method to transform the nonconvex problem into a convex one. However, I cannot see any benefit of transforming Eq.4 into a convex problem. The new learning objective Z in Lemma 1 is just a de-noised version of x_t. The original low-dimensional latent space estimation problem now becomes the problem directly estimating the de-noised high-rank observed state. Thus all merits from low-dimensional assumption are not available now. Since the proposed model is still non-convex, what is its merit compared to classical non-convex matrix factorization style algorithms in LDS?

Reviewer 3



This paper derives a new system identification method that avoids the global optimum method of EM and is more efficient from being MLE based unlike N4SID. The approach is well motivated. The two-view setup in eqn 2 seems to effectively double the observations, i.e. each data point is treated as two data points. In a Bayesian posterior update this would definitely make things go wacky. However, since this method only aims to approximate an MLE point estimate, it may be reasonable. However, what biases this introduces should be explored further. What concerns me more is the replacement of the latent space likelihood P(phi_t|phi_t-1) with the Frobenious norm of the latent states \phi. It is unclear to me why these two penalties should behave alike. What effect this has should be explored. It seems the objective in eqn 4 is more like method that find a joint MAP estimate on the parameters and latent states than method that find an MLE estimate from the marginal likelihood of the data P(X|theta) as EM attempts to do. I would also like to see the regular MSE as a loss function in the table since the predictive mean is not necessarily the Bayes optimal prediction with the NMSE loss function. It is somewhat surprising that EM does much poorer than N4SID in the experiments. How was EM initialized? What happens if one uses EM initialized to the N4SID solution? What about EM initialized to the LDS-MV solution? Comments: The comment on exponential families being well understood on L34 is somewhat vague. Understood in what sense? The \sigma in L248 seems to be the eigenvalues, but this should be explicitly stated. The fonts in Figure 2 are way too small latex: see align package and &=